



# Three-dimensional deformation field analysis of the 2016 Kumamoto Mw 7.1 earthquake

Qingyun Zhang[1], Jingfa Zhang[2], Yongsheng Li[2], Bingquan Li[2], Quancai Xie[3], Sanming Luo[1], Qingzun Ma[1]

[1]The First Monitoring and Application Center, China Earthquake Administration, Tianjin, China

[2]National Institute of Natural Hazards, MEMC, Beijing, China

[3]Key Laboratory of Earthquake Engineering and Engineering Vibration, Institute of Engineering Mechanics, China Earthquake Administration, Harbin, 150080, China

*Correspondence to*: Qingyun Zhang (zhangqingyun1107@163.com)

**Abstract.** The Kumamoto earthquake is analyzed mainly with InSAR data combined with strong earthquake and GPS data, using a variety of joint InSAR technology methods and multisource data solution methods and comprehensively considering the normalization and weighting of multisource data. The three-dimensional (3D) deformation field is determined. The results show that the joint solution of multisource data can improve the accuracy of the 3D solution deformation results to a certain extent. From the 3D solution results, the maximum east-west deformation caused by the 2016 Kumamoto earthquake

is approximately 2 m; the north-south direction mainly manifests expansion and stretching; the northwestern side subsides vertically, with a maximum subsidence of 2 m; and the southeastern side is uplifted. The horizontal deformation characteristics show that the earthquake is dominated by right-lateral strike-slip; the strike is NE-SW, the dip of the seismogenic fault is nearly vertical, and the Futagawa fault has a few normal fault properties. By analyzing the coseismic 3D deformation field, the seismogenic fault can be better understood, which provides a foundation for studying seismic

mechanisms.

## 1 Introduction

At 1:00 on April 16, 2016, Japan time, a magnitude 7.1 earthquake occurred in Kumamoto Prefecture (130.763°E, 32.755°N) in Kyushu, Japan, with a focal depth of 10 kilometers. Before the main shock, there were two foreshocks with Mw ≥ 6.0 in the area, and a series of aftershocks occurred after the main shock. The largest aftershock was Mw 5.7. The Kumamoto

earthquake caused large-scale mountain collapse and damage to houses, roads and bridges. The earthquake caused a rupture of up to 34 km on the surface and a right-lateral displacement of approximately 220 cm along the northern segment of the Butianchuan fault zone.

After the earthquake, the characteristics of the deformation field caused by the earthquake were a general concern and an important reference for subsequent emergency rescue. After the Kumamoto Ms 7.1 earthquake on April 16, some scholars

used interferometric synthetic aperture radar (InSAR) technology to analyze and study the coseismic deformation field, fault parameter inversion, and seismic fault geometry of this earthquake. Fujiwara et al. (2016) and Goto et al. (2017) used InSAR technology in combination with ground surveys to determine exposed fault cliffs and faults caused by the Kumamoto earthquake. Fujiwara et al. (2016, 2017) used the InSAR method to analyze the structural characteristics of seismogenic faults and found that along the northwestern edge of the Aso crater, large deformation was concentrated in the Aso Valley.

Jiang Shengmiao et al. (2018) analyzed the rupture mechanism of the Kumamoto earthquake by combining far-field body wave data and coseismic InSAR data. They determined that right-lateral strike-slip was the main cause of the earthquake rupture and that the rupture may have emerged from the surface. However, a certain degree of ambiguity is introduced as the



ground shape acquired by InSAR is transformed to line-of-sight (LOS) information (Wright et al., 2004). Therefore, direct analysis of deformation field characteristics and source inversion based on the coseismic deformation field from InSAR data provide certain details that are not conducive to the true description of the characteristics of the coseismic deformation field, and the above studies do not involve the acquisition of coseismic three-dimensional (3D) deformation fields.

At present, the research progress of 3D deformation field calculations mainly includes the following features. (1) Direct calculations based on geometric relationships. If three or more LOS deformation fields for the same earthquake can be obtained and the incident and azimuth angles are not exactly the same, the 3D deformation field can be directly calculated according to the equations. The theory of this method is simple and straightforward, but to obtain accurate 3D deformation field results, many observations are needed. Therefore, the same seismic multiview or multisatellite platform data can be used to solve the problem, although the difficult data acquisition and a large amount of calculation are disadvantages (Wright et al, 2004; Gray et al, 2011). At the same time, the incidence angles of SAR satellite sensors launched today are not very different, and SAR data are relatively insensitive to the north-south direction. This factor also leads to relatively low north-south accuracy of the 3D solution results obtained by this method. (2) Joint application of different InSAR technologies. When the acquisition of seismic coseismic deformation data is limited, obtaining SAR data of three or more different perspectives is not possible, but two kinds of data for the same earthquake are obtained from the ascending and descending paths. These data can be considered to combine the LOS deformation field with the distance or azimuth deformation field obtained by other techniques (offset tracking or multi-aperture InSAR (MAI)) for a 3D solution (Michel et al, 1999; Fialko et al, 2001; Funning et al, 2005; Gonzalez et al, 2009; Hu et al, 2010; Gray et al, 2005). The advantage of this method is that the amount of data is relatively low. When acquiring high-resolution SAR image data, the accuracy of the 3D solution can be greatly improved (Jung et al, 2010; Gourmelen et al, 2011). (3) Solution with geodetic data. Differential interferometry can obtain high-resolution planar deformation fields, but only the LOS deformation results can be obtained, and the solution is limited by the SAR satellite revisit period. Geodetic data such as GPS and leveling can provide subcentimeter-level deformation monitoring results. However, due to the constraints of terrain and funding, the GPS points and level control points are unevenly distributed, and the spatial resolution is low. Therefore, some researchers have considered combining the point-type information from GPS and other data with the surface deformation information from SAR data to comprehensively calculate the 3D deformation field (Gudmundsson et al, 2002; Samsonov et al, 2006, 2007; Luo Haibin et al., 2008; Guglioelmino et al., 2011; Catalao et al, 2011; Ban Baosong et al., 2010). (4) 3D solution combined with simulation results. If there are few SAR data resources in the study area, sometimes only one scene of an interference map can be obtained. When there are no other external data for reference constraints, one can consider using the model to simulate the deformation variables and then combining the results with differential interferometry to obtain the optimal 3D deformation field results. However, the simulation process will introduce certain model errors, and the deformation models in different regions will change. Hence, there is no universal deformation model, which causes the method to have certain limitations in the application process.

Based on a synthesis of previous studies and the situation of the Kumamoto seismic data, this paper uses a combination of multiple InSAR techniques and multisource data inversion methods to analyze the 3D deformation field of the Kumamoto earthquake. Through comparative analysis, the joint analysis of multisource data is found to improve the calculation accuracy of the 3D deformation field and provide a reference for further study of source parameters and fault geometric kinematics.

## 2 Geological setting

At 12:26 on April 14, 2016, a magnitude 6.2 earthquake struck near Ikosei-cho, Kamiyoshi-gun, Kumamoto Prefecture, in Kyushu, Japan. The epicenter depth was 11 kilometers, and the temblor killed 9 people, injured hundreds of people, and



collapsed monuments. According to the Japanese intensity scale, the epicenter was estimated as degree 6. The earthquake
occurred approximately 30 kilometers from the center of Aso volcano. The April 16 earthquake struck northeast of the
location of the earthquake on the 14th near the intersection of the Butagawa fault in Ikseiki-cho and the Hinajiu fault. After
the magnitude 7.1 earthquake on April 16, the earthquakes in the previous two days were considered foreshocks, and this
earthquake was the main shock.

The distribution of aftershocks after this earthquake is shown in Figure 1. The seismogenic structure may be a right-lateral
fault striking approximately 30° northeast, most likely due to the Hinagu fault (F2 in Figure 1) and the Futagawa fault (F1 in
Figure 1). The topographic characteristics of these two faults are relatively obvious. The Futagawa fault forms a linear cliff
that rises to the southeast between Aso volcano (red triangle in Figure 1) and Mashiki. The Hinagu fault line is even clearer
between Mashiki and Hinagu Onsen. According to survey results by Japanese scholars at the scene, some faulted surface
ruptures were found in the area of Mashiki. The surface rupture zone is located at 32.797° N and 130.853° E. It generally
shows a trend of 60° north-northeast, with a right-lateral offset of approximately 1.5-2 m and approximately 0.5-0.7 m
vertical displacement. The locations and characteristics of these surface ruptures are mostly consistent with the
Futagawa-Hinagu fault zone mentioned above, so the seismogenic structure of this earthquake was almost certainly the
Futagawa-Hinagu fault system.

## 3 Coseismic deformation field

### 3.1 Strong motion data and GPS coseismic displacement fields

After the earthquake, 1095 free surface strong vibration records of the 365 group of the K-net station network and 1980
strong records of the 990 group of the Kik-net station network in Japan were obtained. This data set contained 990 free
surface data points and 990 downhole array data points. All the data were processed using an improved automatic baseline
correction method (Wang et al., 2011), eliminating the effects of baseline drift, introducing flatness parameters, avoiding
large permanent displacements in the far-field region, and obtaining the east-west, north-south and vertical displacements of
strong motion points. The permanent displacements obtained from the strong motion network records of this earthquake are
shown in Table 2. The coseismic displacement fields obtained are shown by the white arrows in Figure 2. The GPS Earth
Observation Network (GEONET) of Japan obtained measurements of 90 GPS points before and after the earthquake,
estimated the coseismic displacements of the two foreshocks by using the REGARD kinematics (Kawamoto et al., 2016),
compared them with the static displacement extracted through conventional analysis, and finally estimated the coseismic
displacement of the Kumamoto main shock through conventional GEONET analysis. The displacements obtained from the
GPS point records of this earthquake are shown in Table 2. The coseismic displacement fields acquired by GPS are shown by
the blue arrows in Figure 2.

The coseismic displacement field obtained from GPS and strong earthquake data in Figure 2 shows that in the horizontal
direction, several points around the epicenter have very large displacements. Along the fault direction, the northwestern side
of the fault mainly shows deformation in the northeast direction. The southeastern side fault mainly manifests southwest
movement. In the vertical direction, the vicinity of the volcano and the middle of the fault show subsidence, and the upper
left position of the volcano appears to be uplifted.

### 3.2 InSAR data and coseismic deformation field

#### 3.2.1 D-InSAR obtains LOS deformation field



After the Kumamoto earthquake, the ALOS-2 satellite of the Japan Aerospace Exploration Agency (JAXA) acquired SAR data covering the seismic deformation area (see Figure 1). The InSAR information is shown in Table 1. Using the open source software ROI_PAC from the Jet Propulsion Laboratory (JPL)/Caltech and the commercial software Gamma differential interference processing on the ALOS-2 data was performed; a Shuttle Radar Topography Mission (SRTM) digital
elevation model (DEM) with a resolution of 90 m was used to perform terrain removal processing; filtering, phase unwrapping and orbit error correction were then performed. Finally, the seismic coseismic deformation field was obtained by geocoding (Figure 3, where toward the satellite flight direction is positive and away from the satellite flight direction is negative).

Analyzing the results of differential interference reveals that the deformation centers of the InSAR deformation field are
basically distributed in space. There are at least four deformation centers in the entire polar earthquake region. An obvious deformation center at Aso volcano is mainly negative. The upper side of Aso volcano is mostly positive, with a maximum deformation of nearly 2 m, and the vicinity of the central continent is mainly negative, with a maximum deformation of approximately 1.5 m. The central continent near the fault is mainly negative, with a maximum deformation of approximately 1.5 m, and the negative area shown by the ascending orbit is larger than the area shown by the descending orbit. At the
junction of the Hinagu fault and the Futagawa fault, the deformation field is positive on the northwestern side of the fault and negative on the southeastern side. The deformation field indicates that the fault induced by this earthquake is not a simple single fault but may be the result of the combined actions of the Hinagu fault and the Futagawa fault. The red square and circle points in Figure 3 are the data points used for verification in the subsequent 3D deformation field solution.

### 3.2.2 Using MAI to obtain the azimuth deformation field

The MAI technology (Bechor et al., 2006) is an InSAR processing algorithm that uses the plus and minus values of the radar beam Doppler shift and splits the beam to obtain the azimuth deformation information for the target. The two single look complex (SLC) images are split into two pairs of front-view and back-view master-slave images according to the Doppler frequency shift sign, and these pairs interfere separately. Because the spatial baselines of the front-view and back-view interferograms are almost the same, they receive the same atmospheric images, and the distance deformations they detect are
also the same. Therefore, the phase difference between the front-view and back-view interferograms is used to obtain the phase information for the azimuth deformation.

Figure 4 shows the results for the azimuth deformation obtained using the MAI method. The Kumamoto earthquake caused severe azimuth deformation, which is mainly manifested by more than two deformation centers. The fault demarcation line is obvious. The seismogenic fault is not a single fault line. It may be a double fault or even a three-fault model. The maximum
deformation amount in the azimuth direction is 1 m. The azimuth deformation obtained in the ascending orbit data has some orbital errors, and the disc deformation amount under the fault is large. Moreover, the amount of deformation in the southeastern region of the fault is large, while the amount of deformation in the northwestern region is relatively small. Both ends of the northwestern region of the fault are positive, but the middle is negative. The azimuth deformation field of the descending orbit data is relatively complete, and the forms of the azimuth deformation fields from the ascending and
descending orbit are opposite, which is mainly caused by the SAR sensor imaging mode. The azimuth deformation from the descending orbit on the southeastern side of the fault is positive, while that on the northwestern side is negative, and the deformation characteristics at both ends of the fault are obvious. The results of the azimuth deformation field illustrate that due to the influence of the satellite flight direction and the SAR imaging mechanism, there are some differences in the deformation characteristics of the azimuth deformation field obtained from ascending and descending orbit data. The regions
with negative values in ascending orbit data have positive values in descending orbit data, but the distribution of the overall deformation center is highly consistent, and the deformation magnitudes are basically the same.


## 4 Three-dimensional deformation field acquisition

Both the LOS deformation field and the azimuth deformation field can be regarded as sets of east-west, north-south, and vertical components. The three-direction deformation variables contribute differently to the LOS deformation field and the azimuth deformation field. The SAR imaging geometry (Figure 5) shows that the LOS deformation is positive when approaching the radar direction and negative when the movement is away from the radar direction. The north, east, and up directions in the geographic coordinate system are specified as positive values, and the LOS deformation $D_{LOS}$ and azimuth deformation $D_{AZI}$ can be expressed as:

$$d_{LOS} = d_U \cos\theta - d_N \sin\theta \cos\left(\alpha - \frac{3\pi}{2}\right) - d_E \sin\theta \sin\left(\alpha - \frac{3\pi}{2}\right) \tag{1}$$

$$d_{AZI} = d_N \sin\left(\alpha - \frac{3\pi}{2}\right) - d_E \cos\left(\alpha - \frac{3\pi}{2}\right) \tag{2}$$

In the equations, $d_U$, $d_E$ and $d_N$ represent the vertical, east-west, and north-south deformation variables, respectively; $\theta$ is the radar incidence angle, $\alpha$ is the angle between the clockwise direction of the satellite flight direction and the north direction, and $\alpha - \frac{3}{2}\pi$ is the distance direction and the positive direction. The angle between the north direction and the clockwise direction is also called the azimuth look direction (ALD).

Based on the Kumamoto earthquake data acquired in Japan, we consider using a combination of multiple InSAR technologies (differential InSAR (D-InSAR) combined with MAI technology) and multisource data (D-InSAR, GPS, and strong motion data) for a 3D deformation field solution. The basic principles of these two methods are introduced below.

### 4.1 Three-dimensional deformation field calculation principle based on a combination of multiple InSAR technologies

If the ascending and descending orbit data of the earthquake can be obtained, MAI processing can be performed on these data to obtain the azimuth deformation results, and differential interference processing can be performed to obtain the LOS deformation results. Given the relationship between the LOS deformation results and the azimuth deformation results, the true 3D deformation results on the ground surface are shown in Figure 5. Based on the data, the following four equations can be obtained:

$$
\begin{aligned}
d_{LOS}^{A} &= d_U \cos\theta^A - d_N \sin\theta^A \cos\left(\alpha^A - \frac{3\pi}{2}\right) - d_E \sin\theta^A \sin\left(\alpha^A - \frac{3\pi}{2}\right) \\
d_{AZI}^{A} &= d_N \sin\left(\alpha^A - \frac{3\pi}{2}\right) - d_E \cos\left(\alpha^A - \frac{3\pi}{2}\right) \\
d_{LOS}^{D} &= d_U \cos\theta^D - d_N \sin\theta^D \cos\left(\alpha^D - \frac{3\pi}{2}\right) - d_E \sin\theta^D \sin\left(\alpha^D - \frac{3\pi}{2}\right) \\
d_{AZI}^{D} &= d_N \sin\left(\alpha^D - \frac{3\pi}{2}\right) - d_E \cos\left(\alpha^D - \frac{3\pi}{2}\right)
\end{aligned}
\tag{3}
$$

The above equations are converted to a matrix form, which yields the following:

$$L = BX \tag{4}$$


$$B = \begin{bmatrix} \cos\theta^A & -\sin\theta^A\cos\left(\alpha^A - \dfrac{3\pi}{2}\right) & -\sin\theta^A\sin\left(\alpha^A - \dfrac{3\pi}{2}\right) \\ 0 & -\sin\left(\alpha^A - \dfrac{3\pi}{2}\right) & \cos\left(\alpha^A - \dfrac{3\pi}{2}\right) \\ \cos\theta^D & -\sin\theta^D\cos\left(\alpha^D - \dfrac{3\pi}{2}\right) & -\sin\theta^D\sin\left(\alpha^D - \dfrac{3\pi}{2}\right) \\ 0 & -\sin\left(\alpha^D - \dfrac{3\pi}{2}\right) & \cos\left(\alpha^D - \dfrac{3\pi}{2}\right) \end{bmatrix}$$

( 5 )

$$L = \begin{bmatrix} d_{LOS}^A & d_{AZI}^A & d_{LOS}^D & d_{AZI}^D \end{bmatrix}^T \quad X = \begin{bmatrix} d_U & d_N & d_E \end{bmatrix}^T$$

( 6 )

According to the principle of least squares, considering the influence of measured values on noise during the process of differential interference processing, the optimal solution of the 3D surface deformation field can be obtained:

$$X = \left(B^T P B\right)^{-1} B^T P L$$

( 7 )

where $P$ is a weight matrix composed of the variance of observations, which can be obtained based on the deformation measurement results.

Using the azimuth deformation field and the LOS deformation field, according to the abovementioned least-squares solution method, the obtained deformation field is transformed into the same coordinate system and sampled to the same resolution, and the standard deviation of the deformation field is then calculated. Finally, the 3D deformation field is solved with Eq. 7.

### 4.2 Solution of the 3D deformation field of Multisource data based on least squares

Although the azimuth and LOS fusion method can be used to solve the 3D deformation field, the accuracies of the LOS deformation field and the azimuth deformation field are affected by the accuracy of error removal during the SAR data processing. Therefore, GPS and strong earthquake data are considered for calculation. However, the spatial references and scales of GPS and strong motion data are different from those of SAR data, resulting in errors in the fusion between point data and InSAR surface data. The main sources of error are as follows:

1) Time scale error. Continuous coseismic GPS points can be obtained from the data for several hours or days before and after the earthquake. However, the InSAR data are affected by the satellite revisit cycle and depend on the situation. The ALOS-2 satellite revisit cycle is 14 days for the images used in this article. Therefore, the time intervals of the acquired data are different for the same earthquake. Since a large earthquake is accompanied by many aftershocks, the superposition of aftershocks also causes crustal deformation, which leads to diverse deformation monitoring results at different time scales.

2) Observational error. Data from different sources have their associated observational errors. A GPS observation station can obtain the 3D deformation information for the station within a certain period of time, but the accuracy of the obtained vertical deformation is lower than that of the horizontal deformation. Affected by the SAR satellite flight orbit, the LOS deformation results obtained from InSAR data are not sensitive to north-south deformation. Therefore, when performing joint analysis of multisource data, the influence of observational errors needs to be considered.

3) The problem of spatial scale. Both the GPS and strong motion data yield the deformation variables of the site within a certain period of time. After the SAR data are processed, the deformation variables of the image pixels are obtained. In the process of SAR imaging, several effects such as overlay, shadow and dihedral angle reflection lead to some differences in spatial position between the deformation field obtained by SAR and the deformation field obtained by GPS and strong earthquake data, thus causing spatial scale errors.

4) Different reference frames. The reference coordinate systems of GPS stations and strong motion points are generally Eurasian reference coordinate systems, such as WGS84. The reference point in SAR differential interferometry is selected in

the radar coordinate system. Two different reference coordinate systems also affect the deformation field results to some extent.

Therefore, when introducing GPS and strong motion data for 3D calculations, it is necessary to fully consider these error sources and deal with various errors appropriately to reduce the impacts of errors on the results. Because the deformation field obtained by GPS and strong earthquake data is converted to the radar line of sight, the deformation trend is consistent

with the deformation field obtained by differential interferometry. Therefore, after normalization, the 3D deformation field solution of multisource data can be realized.

The most critical step of the 3D solution method using multisource data based on least squares is the normalization of different source data and the problem of data weighting. According to the error sources of the different source data introduced above, the relevant errors are removed, and the 3D deformation field is then solved. The specific process is as

follows:

1) To avoid the influence of time error, the GPS and strong motion data are chosen at the same time point before and after the earthquake for processing to obtain a point-shaped coseismic 3D deformation field. At the same time, the SAR data closest to the time of the earthquake are selected for differential interference processing and MAI processing to obtain the LOS deformation and the azimuth deformation, respectively.

2) The GPS points and strong motion points are converted to the WGS84 coordinate system. The coseismic deformation field obtained from SAR data processing is also converted to the WGS84 coordinate system while ensuring that the results of the ascending and descending orbits have the same resolution.

3) Regression analysis is performed on different source data for reasons such as inconsistent starting points. Since the distribution densities of GPS and strong motion points are relatively low, GPS data and strong motion data are first fused.

Both the GPS deformation field and the deformation field obtained from strong motion data provide 3D deformation information for a point source, so the regression analysis method can be used for fitting, and the data can be unified and normalized to increase the density of the point deformation field.

4) Multisource data weighting. Considering the reference points and inconsistent scales of different source data, the weights of the normalized multisource data are determined. An analysis of the weighting methods, such as the comparison

method and empirical method, indicates that these methods are too subjective; finally, the deformation observation measurement variance is selected to determine the weights. The three-component results of the fused GPS and strong motion data are converted into the LOS deformation results.

$$G_{LOS} = \begin{bmatrix} d_U^{fusion} & d_E^{fusion} & d_N^{fusion} \end{bmatrix} \begin{bmatrix} \cos\theta & -\sin\theta\sin\alpha & -\sin\theta\cos\alpha \end{bmatrix}^T$$

( 8 )

According to the standard deviation equation of double difference observations, the observed variance of the LOS

deformation results obtained for the fusion site and the corresponding differential interference, are calculated as shown in Eq. 9:

$$\sigma_{ins,fusion}^2 = \frac{1}{2n}\sum_{i=1}^{n}\left(d_{fusion,i}^{los} - d_{ins,i}^{los}\right)^2$$

( 9 )

where $n$ represents the number of sites after fusion; $\sigma_{ins,fusion}^2$ represents the differential interferometry variance corresponding to the site after fusion; $d_{fusion,i}^{los}$ represents the deformation variable converted from the 3D deformation

observation measurement of the $i$ th site to the LOS direction; and $d_{ins,i}^{los}$ is the deformation variable of the differential interference deformation field corresponding to the $i$ th site.

The cell variance is calculated with Eq. 10:





$$\sigma_{ins}^2 = \frac{E\left(\gamma_{fusion}\right)}{\gamma_i} \sigma_{ins,fusion}^2$$

(10)

where $\gamma_i$ is the coherence coefficient of the $i$th pixel and $E\left(\gamma_{fusion}\right)$ is the mean of the coherence coefficient of the SAR image pixel corresponding to the fused site.

Although the GPS and strong motion data are fused, they remain point data after fusion, and evaluating the accuracy of the planar LOS deformation field is impossible. Considering the effects of orbital errors and terrain fluctuations, a model is established between pixel variance and pixel position and elevation, and the accuracy of the pixels obtained by differential interferometry is evaluated. The equation is as follows:

$$\sigma_{ins,i}^2 = a_0 + a_1 x + a_2 y + a_3 x^2 + a_4 xy + a_5 y^2 + b_1 z + b_2 z^2$$

(11)

The first six terms on the right side of the equation are the orbital error terms. $x$ is the azimuth coordinate of the $i$th pixel, and $y$ is the distance coordinate of the $i$th pixel. The last two terms describe terrain errors, where $z$ represents the elevation of cell $(x, y)$. $a_i\left(i = 1, 2, \cdots, 5\right)$ and $b_j\left(j = 1, 2\right)$ are the parameters to be calculated.

The coefficient parameters are calculated according to Eq. 11, and the standard deviation $\sigma_{ins,i}^{LOS}$ of each pixel in the LOS deformation field obtained by SAR is calculated through the fitted model.

Given the LOS deformation field result variance and azimuth deformation result variance $\sigma_{ins,i}^{LOS}$ and $\sigma_{ins,i}^{AZI}$, respectively, which are obtained from the point where GPS and strong motion data are fused, the weights are then determined according to Eq. 12:

$$P = \left(diag\left(\left(\sigma_{ins,i}^{LOS,1}\right)^2, \left(\sigma_{ins,i}^{LOS,2}\right)^2, \left(\sigma_{ins,i}^{AZI,1}\right)^2, \left(\sigma_{ins,i}^{AZI,2}\right)^2\right)\right)^{-1}$$

(12)

where $\left(\sigma_{ins,i}^{LOS,1}\right)^2, \left(\sigma_{ins,i}^{LOS,2}\right)^2, \left(\sigma_{ins,i}^{AZI,1}\right)^2, \left(\sigma_{ins,i}^{AZI,2}\right)^2$ represent the variances of the ascending and descending data.

5) 3D deformation field solution. After calculating the weight matrix, the 3D surface deformation field is calculated based on the principle of least squares.

### 4.3 Solution of the 3D deformation field of the Kumamoto earthquake in Japan

The LOS orientation, azimuth deformation, GPS and strong motion data are combined to solve the 3D deformation field of the Kumamoto earthquake. The 3D coseismic deformation field calculations are performed using a combination of multiple InSAR technologies and a multisource data fusion method. The results are shown in Figure 6.

In Figure 6, the first line (Figure 6 (a, b, c)) shows the result of 3D deformation field calculation using a variety of InSAR technologies; the second line (Figure 6 (d, e, f)) shows the use of multisource data fusion results for the 3D deformation field solution; the first column (Figure 6 (a, d)) represents east-west deformation; the second column (Figure 6 (b, e)) represents north-south deformation; and the third column (Figure 6 (c, f)) shows the vertical deformation. The overall deformation field illustrates that the deformation characteristics are clearly delimited along the fault line. In the east-west deformation field, the Aso volcanic region and the southeastern side of the fault mainly show westward movement, and the northwestern side of the fault mainly shows eastward movement. The east-west deformation field jointly obtained by InSAR technology is larger in deformation level than the result obtained by multisource data fusion. Comparing the black boxes in the deformation fields in Figure 6 (a) and (d) shows anomalous jumps in the east-west deformation field obtained by using a combination of multiple InSAR techniques, and the overall deformation field is not continuous. In the north-south direction, the



northwestern side of the fault mainly moves northward, and the southeastern side of the fault moves southward. The maximum amount of motion is nearly 2 m. The results obtained by this method are basically the same. The maximum deformation value in Figure 6 (b) is slightly larger than the result in Figure 6 (e), which may be caused by the relative
insensitivity of the SAR image in the north-south direction. The north-south deformation shows obvious dextral strike-slip properties. The vertical direction mainly shows downward movement, especially in the Aso volcanic area and the middle area of the fault. Both ends of the fault and the southeastern side of the fault show uplift. The results obtained by the two methods are consistent. The 3D deformation calculation results reveal that the seismogenic structure of this earthquake is relatively complicated, but the main feature of the fault is the dextral strike-slip movement.

**4.4 Results and analysis**

To better compare the accuracy of the two 3D deformation field calculation methods, the GPS and strong motion data distributed in the overlapping area of SAR ascending and descending orbit data are selected. The deformation results obtained by the GPS and strong motion data in this area are normalized. The GPS and strong motion data in the overlapping area are shown in Table 2. Among them, the site names starting with K are the strong motion data. The numerically named
points are the GPS data, and the results are compared with the 3D solution results (Figure 7).

Figure 7 shows the comparison between the 3D solution results and the GPS and strong motion data results. The deformation field results obtained by GPS and strong motion data are compared with the 3D solution results obtained by the two methods, and a graph is then drawn. The closer the curve is to 0, the higher the agreement. The blue curve shows the comparison between the results from the combination of multiple InSAR technologies and the GPS and strong motion data. The yellow
curve shows the comparison of the GPS and strong motion data results from the 3D solution of multisource data fusion.

Figure 7 (a) shows a comparison of the east-west deformation field results, which indicates that the difference between the 3D solution results from multisource data fusion and the GPS and strong motion data results is close to zero. The difference curve between the 3D solution obtained by multiple InSAR technologies and the GPS and strong motion data fluctuates greatly, especially at points 2, 9, and 19. Although the differences are relatively large within ±0.8 cm, the results of the joint
solution of multisource data are more stable. Figure 7 (b) shows the comparison of the north-south deformation field results. These results reveal that the difference curves at points 3, 5, and 7 with GPS and strong motion data in the 3D solution results obtained by a combination of multiple InSAR technologies fluctuate greatly. The results of the joint solution of multisource data are relatively more stable and basically fluctuate within ±0.2 cm. Figure 7 (c) shows a comparison of the vertical deformation results. The difference curves between points 2 and 4 and the GPS and strong motion data results in the
3D solution obtained by multiple InSAR technologies are undulating, and the remaining points are stable within ±0.5 cm. The results of the joint calculation of multisource data also have relatively large fluctuations at points 2 and 4, but overall, they are relatively stable.

A comparison of the 3D solution results with the GPS and strong motion data results shows that the 3D deformation field based on the combination of multisource data is relatively more reliable and that the 3D deformation field results obtained
are more accurate. Moreover, the problem of SAR deformation field insensitivity in the north-south direction can be overcome to a certain extent. However, affected by factors such as the location of the earthquake zone, many earthquake areas do not have sufficient densities of GPS or strong earthquake observation points. To date, using a variety of InSAR technologies can also yield relatively reliable 3D deformation field calculation results.

The coseismic 3D deformation field (Figure 8) in the near-field region shows that the 2016 Kumamoto earthquake was
dominated by NE-SW right-lateral strike-slip. The strike-slip magnitudes of the north and south sides are equal, and the relative displacements are almost the same, indicating that the fault dip is nearly vertical. There is a significant difference in the movements of the horizontal displacement component on either side of the seismogenic fault. The northwestern side





moves to the northeast, and the southeastern side moves to the southwest, indicating that the 2016 Kumamoto earthquake was a NE-SW strike-slip earthquake. The northwestern part of the Futagawa fault is mainly characterized by surface subsidence, and the southeastern part is slightly uplifted, indicating that the Futagawa fault is a right-lateral strike-slip fault with normal fault characteristics.

The analysis of the deformation field profile (Figure 9) shows that the largest deformation of this earthquake occurred near the fault. The three profile lines reflect typical coseismic deformation characteristics. The deformations on both sides of the fault are relatively continuous, and along the surface of the fault, obvious dislocation occurred. Section A-A' is located in the southwestern part of the fault, and the deformation is approximately symmetrically distributed around the fault itself; the northwestern side is mainly characterized by surface uplift, and the southern side manifests surface subsidence. The near-fault area in section B-B' in the middle of the fault shows land subsidence, with a maximum subsidence of nearly 2 m. The subsidence of the northwestern side is much larger than the uplift of the southeastern side, indicating that the surface of the near-fault area is dominated by subsidence; the deformation is mainly concentrated within 10 km of the fault, and there are faster attenuation rates on both sides of the fault. Section C-C' is located in the northeastern part of the fault, and the horizontal displacements on both sides of the fault are greater than the vertical displacements. This deformation feature is consistent with the nature of this earthquake dominated by strike-slip motion. The east-west and north-south deformations along profiles B-B' and C-C' are relatively consistent, and there are obvious dislocations and ruptures on both sides of the fault. The largest north-south deformation caused by this earthquake occurred mainly near the fault and attenuated rapidly in the direction away from the fault. This earthquake appears to have been an obvious right-lateral strike-slip earthquake.

## 5 Conclusion

The acquisition of a coseismic 3D deformation field holds great significance for an in-depth understanding of seismic deformation characteristics and the analysis of seismogenic fault properties. Taking the 2016 Kumamoto earthquake in Japan as an example, this paper uses a variety of InSAR technology methods and a joint multisource data solution method to solve the 3D deformation field, considering the data normalization and weight determination in the latter method. This approach successfully obtains the coseismic 3D deformation field of the Kumamoto earthquake. The study finds that the quality of the 3D deformation field obtained after adding GPS and strong earthquake data constraints is significantly improved, indicating that the joint solution of multisource data can yield a reliable coseismic 3D deformation field. The analysis of the 3D deformation results shows that the main seismogenic fault of this earthquake is a right-lateral strike-slip seismogenic system. Among the faults, the Futagawa fault also has a small amount of normal slip. The seismic zone strikes NE-SW, and the dip of the seismogenic fault is almost vertical. The surface of the northwestern side obviously subsides in the middle of the fault, the northeastern and southwestern ends of the fault appear to be uplifted, and the southeastern side has an uplift trend along the fault zone, which is consistent with the main nature of the right-lateral strike-slip motion for this earthquake. The deformation caused by the earthquake is mainly manifested as expansive stretching in the north-south direction; in the east-west direction, the northwestern side moves eastward, with a maximum deformation of nearly 2 m; vertically, the maximum subsidence of the northwestern side is approximately 2 m, and the maximum uplift of the southeastern side is 0.6 m. The deformation field results obtained by the 3D solution agree well with the GPS and strong earthquake station data.

## Data availability

The ALOS-2 data were obtained from the Dragon IV program (10607).





## Author contributions

QZ designed the general idea and wrote the text content, JZ contributed to paper writing and revision, YL and BL was the technical director in chief, QX processed GPS and strong motion station data, SL and QM contributed to paper writing. All authors discussed the editors' opinions and revised the paper.

## Competing interests

The authors declare that they have no conflict of interest.

## Financial support

This research has been supported by Tianjin Natural Science Foundation (grant no. 20JCQNJC01360).

## Acknowledgments

The Dragon Plan provided the PALSAR-2 data from the Japanese Aerospace Exploration Agency (JAXA), in the mid-term period of the fourth phase. The Generic Mapping Tools (GMT) software was used for mapping.

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

**Figures and Tables:**

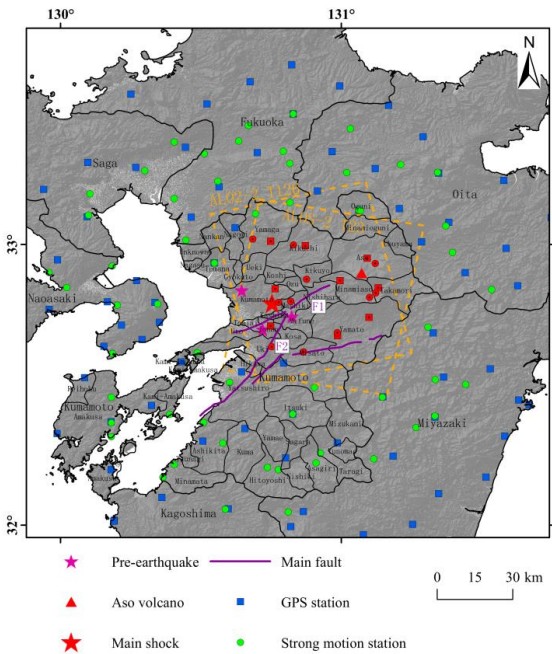

**Fig. 1 Regional tectonic setting of the Kumamoto earthquake**

The red pentagram indicates the location of the main shock; the purple lines are fault lines, where F1 represents the Futagawa fault and F2 represents the Hinagu fault; the yellow dashed box shows the coverage of the SAR data used. Terrain is from the SRTM produced by NASA.

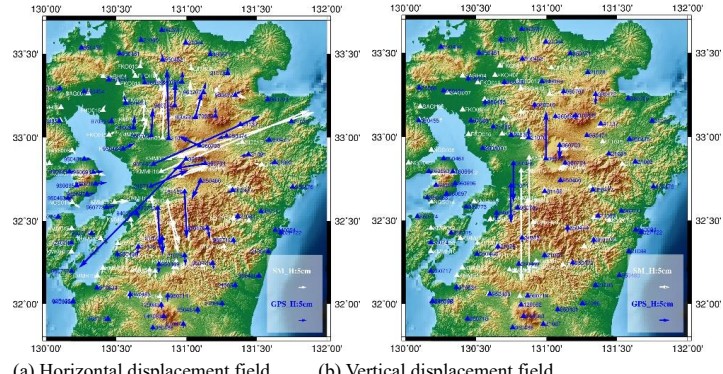

(a) Horizontal displacement field       (b) Vertical displacement field

**Fig. 2 Coseismic displacement fields acquired by GPS and strong motion data on April 16, 2016. Terrain is from the SRTM produced by NASA.**





**Table 1 InSAR data used for Kumamoto earthquake research**

| Satellite type | Orbit number | Flight direction | Imaging mode | Wavelength/(cm) | Master image | Slave image | Time baseline/(day) | Incidence angle/(°) |
|---|---|---|---|---|---|---|---|---|
| ALOS-2 | 126 | Ascending | Stripe | 24.245 | 20160415 | 20160429 | 14 | 36 |
| ALOS-2 | 23 | Descending | Stripe | 24.245 | 20160307 | 20160418 | 42 | 36 |

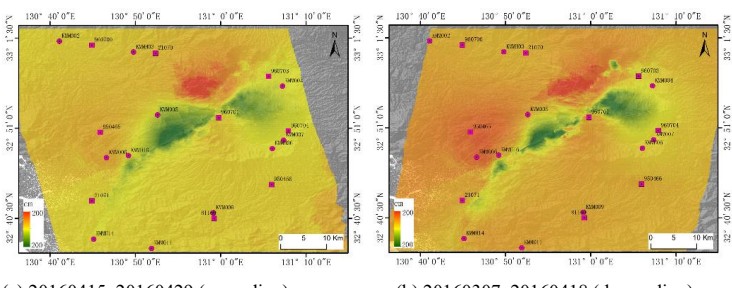

(a) 20160415_20160429 (ascending)      (b) 20160307_20160418 (descending)

**Fig. 3 InSAR coseismic deformation fields**

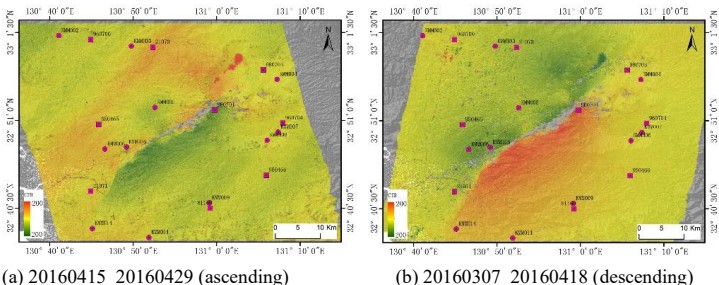

(a) 20160415_20160429 (ascending)      (b) 20160307_20160418 (descending)

**Fig. 4 Azimuthal deformation fields obtained from ALOS-2 for the Mw 7.1 Kumamoto earthquake on April 16, 2016.**

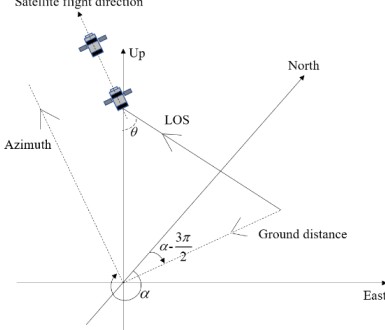

**Fig. 5 SAR imaging geometry. The positive directions of the coordinate system in the figure are the directions of the arrows.**





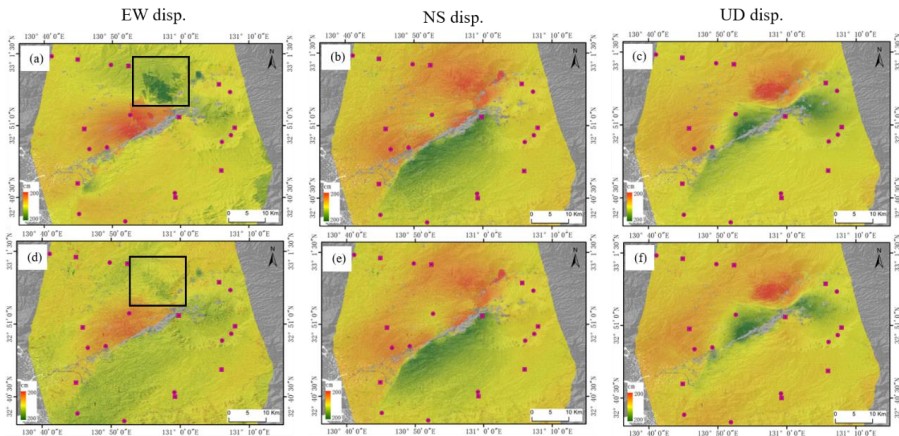

**Fig. 6 3D deformation field of the Mw 7.1 earthquake in Kumamoto on April 16, 2016.**

**(a) ~ (c) Multiple InSAR technology joint method; (d) ~ (f) Multisource data joint method; (a) and (d) E-W horizontal displacement; (b) and (e) N-S horizontal displacement; (c) and (f) U-D (vertical) displacement; positive signals indicate eastward, northward and upward movements.**


**Table 2 Three-component deformation results obtained from strong motion and GPS data**

| Number | Site name | Longitude (°) | latitude (°) | Epicentral distance (km) | EW (cm) | NS (cm) | UD (cm) |
|---|---|---|---|---|---|---|---|
| 1 | KMM006 | 130.777 | 32.793 | 4.500 | 72.120 | 50.810 | -24.230 |
| 2 | KMMH16 | 130.820 | 32.797 | 7.100 | 95.610 | 39.550 | -55.490 |
| 3 | KMMH14 | 130.752 | 32.635 | 13.400 | 4.827 | -24.550 | -2.909 |
| 4 | KMM005 | 130.877 | 32.876 | 17.200 | 93.720 | 29.700 | -58.000 |
| 5 | KMM011 | 130.865 | 32.617 | 18.100 | 6.711 | -33.540 | -3.848 |
| 6 | KMM009 | 130.986 | 32.686 | 22.200 | 2.653 | -33.090 | -4.319 |
| 7 | KMMH03 | 130.830 | 32.998 | 27.800 | 1.109 | 36.180 | 2.843 |
| 8 | KMM002 | 130.685 | 33.019 | 30.300 | 3.286 | 11.220 | 1.692 |
| 9 | KMMH06 | 131.101 | 32.811 | 32.200 | -16.470 | -5.650 | 1.475 |
| 10 | KMM007 | 131.123 | 32.827 | 34.600 | -18.470 | -5.491 | 0.531 |
| 11 | KMM004 | 131.121 | 32.932 | 38.900 | -17.880 | -6.051 | -0.858 |
| 12 | 21071 | 130.748 | 32.709 | 5.216 | 24.211 | 10.360 | -18.879 |
| 13 | 950465 | 130.765 | 32.842 | 9.742 | 68.892 | 30.852 | -19.454 |
| 14 | 81169 | 130.987 | 32.675 | 22.751 | 1.349 | -30.279 | -0.613 |
| 15 | 960701 | 130.996 | 32.871 | 25.340 | -69.161 | -70.029 | 23.756 |
| 16 | 960700 | 130.749 | 33.011 | 28.586 | 1.034 | 18.114 | 0.739 |
| 17 | 21070 | 130.873 | 32.996 | 28.734 | -1.143 | 44.627 | 5.028 |
| 18 | 950466 | 131.099 | 32.741 | 31.490 | -5.750 | -11.951 | -0.288 |
| 19 | 960704 | 131.132 | 32.845 | 35.942 | -20.680 | -3.429 | -1.513 |
| 20 | 960703 | 131.093 | 32.951 | 37.795 | -13.447 | 6.043 | -7.351 |


(a) E-W horizontal displacement


(b) N-S horizontal displacement

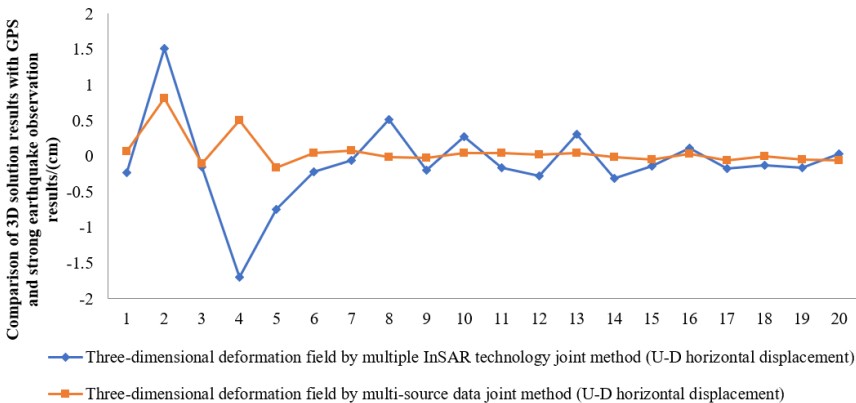

(c) U-D horizontal displacement

**Fig. 7 Comparison of 3D solution results with GPS and strong earthquake observation results for the Mw 7.1 earthquake in**
**Kumamoto on April 16, 2016.**

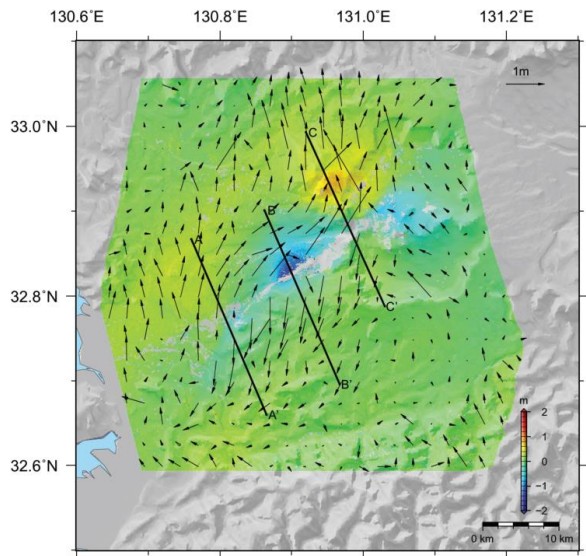

**Fig. 8 Near-field 3D deformation field of the Kumamoto earthquake by multisource data joint method.**

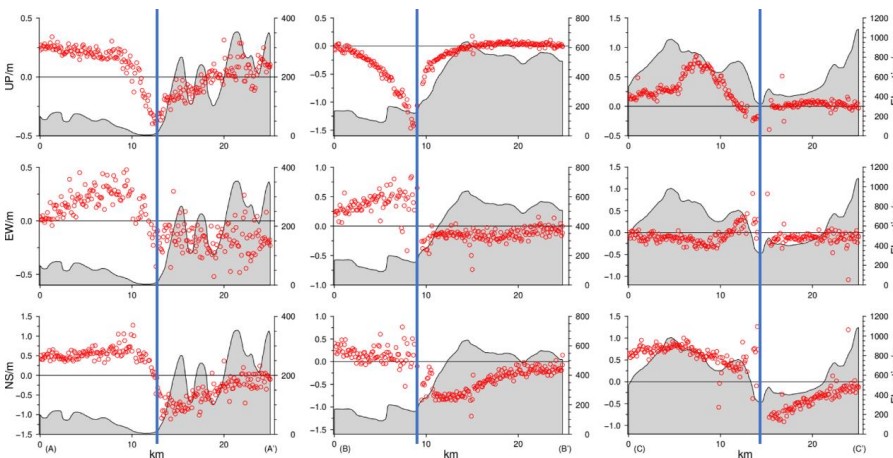


**Fig. 9 A-A' (left), B-B' (middle) and C-C' (right) deformation profiles. The blue lines mark the fault location.**