# Peer review of "Three-dimensional deformation field analysis of the 2016 Kumamoto"

_Natural Hazards and Earth System Sciences, 2020_

## Referee Comment (RC1)

The authors used regular InSAR and MAI results from ALOS-2, GPS, and the strong earthquake data to estimate the 3D coseismic deformation field of the 2016 Kumamoto earthquake, and compared with using InSAR results (InSAR + MAI) only, the authors found they can improve the 3D displacements field by fusing external GPS and strong earthquake data, which will be helpful for studying the source models of the earthquake. Technically, the presented methodology has been widely used in the previous researches, so the authors need to weaken their words about the description of "innovation".

Several other comments:

(1) About equation 7, the authors should clarify how they calculate the weights. Only by modeling the decorrelation noise based on the coherence? But decorrelation noise is not the major error source of InSAR in many cases.

(2) About equation 11, did the authors want to fit the variance? To my understand, the obit errors can be modeled in space, but not for their variances.

(3) About equation (12), how to estimate the variance of MAI observations? Please clarify it.

(4) Line 350-355: "The study finds that the quality of the 3D deformation field obtained after adding GPS and strong earthquake data constraints is significantly improved". It would be good to present solid evidence about the "significant improvement".

---

## Community Comment (CC2)

**We appreciate the thorough review of our manuscript and the constructive feedback provided by Sylvain Barbot. We have addressed the comments point by point and made revisions accordingly as follows.**

**Reply to community comment**

*1. The paper "Three-dimensional deformation field analysis of the 2016 Kumamoto Mw 7.1 earthquake" by Zhang et al. provides a technical, straightforward methodology to combine either multiple InSAR data or heterogeneous InSAR and geodetic datasets to build a 3-component displacement map for earthquakes. The technique is well known, being used for almost two decades. The paper is technically correct, but its novelty is questionable.*

**Author's reply:**

This paper is not a simple application for estimating the three-dimensional deformation field but a comparative study. The case of the 2016 Kumamoto earthquake is to analyze different InSAR technologies, multi-source data, and their application for the estimation. Through the comparative analysis, the application of multi-source data can improve the accuracy for estimating the three-dimensional deformation field. This paper also provides a reference for the subsequent application cases for estimating the three-dimensional deformation field.

*2. Line 55, azimuthal InSAR is also described in*
*Barbot, S., Hamiel, Y. and Fialko, Y., 2008. Space geodetic investigation of the coseismic and postseismic deformation due to the 2003 Mw7. 2 Altai earthquake: Implications for the local lithospheric rheology. Journal of Geophysical Research: Solid Earth, 113(B3).*

**Author's reply:**

Thank you very much for your insightful comment and presenting us relevant literature. We think the reference on the azimuthal InSAR is very important for improve the credibility of the manuscript. It had been cited in the revised manuscript. The sentence had been changed to "These data can be considered to combine the LOS deformation field with the distance or azimuth deformation fieldobtained by other InSAR techniques (e.g., Barbot et al., 2008; Bechor et al., 2006; Michel et al., 1999) for a 3D solution (e.g., Fialko et al, 2001; Funning et al, 2005; Gonzalez et al, 2009; Hu et al, 2010; Gray et al, 2005)."

*3. Line 356: I do not see a justification for the vertical fault. Modeling of the deformation indicates north-dipping faults. See*
*Moore, J.D., Yu, H., Tang, C.H., Wang, T., Barbot, S., Peng, D., Masuti, S., Dauwels, J., Hsu, Y.J., Lambert, V. and Nanjundiah, P., 2017. Imaging the distribution of transient viscosity after the 2016 Mw 7.1 Kumamoto earthquake. Science, 356(6334), pp.163-167.*

**Author's reply:**

Thank you very much for pointing out the incorrect description on the dip of fault. The dip in the

paper (Moore et al., 2017) are 69 and 75 degrees. The description on the fault in the revised manuscript had been changed to "The strike-slip magnitudes of the north and south sides are equal, and the relative displacements are almost the same, indicating that the fault dip maybe big. Under the guidance of seismic depth distribution, Moore et al. (2017) obtained the dip angles of the seismogenic fault by trial and error method using two faults for coseismic slip inversion, and the dip angle of the northeast segment is 69 degrees, and that of the southwest segment is 75 degrees."